# Estimation of Aboveground Biomass for Winter Wheat at the Later Growth Stage by Combining Digital Texture and Spectral Analysis

**Ling Zheng [1,\*], Qun Chen [1], Jianpeng Tao [1], Yakun Zhang [2], Yu Lei [1], Jinling Zhao [1] and Linsheng Huang [1]**

[1] National Engineering Research Venter for Agro-Ecological Big Data Analysis & Application, Anhui University, Hefei 230601, China

[2] College of Agricultural Equipment Engineering, Henan University of Science and Technology, Luoyang 471003, China

[\*] Correspondence: lingz0865@163.com

**Abstract:** Aboveground biomass (AGB) is an important indicator used to predict crop yield. Traditional spectral features or image textures have been proposed to estimate the AGB of crops, but they perform poorly at high biomass levels. This study thus evaluated the ability of spectral features, image textures, and their combinations to estimate winter wheat AGB. Spectral features were obtained from the wheat canopy reflectance spectra at 400–1000 nm, including original wavelengths and seven vegetation indices. Effective wavelengths (EWs) were screened through use of the successive projection algorithm, and the optimal vegetation index was selected by correlation analysis. Image texture features, including texture features and the normalized difference texture index, were extracted using gray level co-occurrence matrices. Effective variables, including the optimal texture subset (OTEXS) and optimal normalized difference texture index subset (ONDTIS), were selected by the ranking of feature importance using the random forest (RF) algorithm. Linear regression (LR), partial least squares regression (PLS), and RF were established to evaluate the relationship between each calculated feature and AGB. Results demonstrate that the ONDTIS with PLS based on the validation datasets exhibited better performance in estimating AGB for the post-seedling stage ($R^2 = 0.75$, RMSE = 0.04). Moreover, the combinations of the OTEXS and EWs exhibited the highest prediction accuracy for the seeding stage when based on the PLS model ($R^2 = 0.94$, RMSE = 0.01), the post-seedling stage when based on the LR model ($R^2 = 0.78$, RMSE = 0.05), and for all stages when based on the RF model ($R^2 = 0.87$, RMSE = 0.05). Hence, the combined use of spectral and image textures can effectively improve the accuracy of AGB estimation, especially at the post-seedling stage.

**Keywords:** aboveground biomass; wheat; canopy; vegetation indices; texture

## 1. Introduction

Aboveground biomass (AGB) is of great practical significance to the monitoring of crop growth [1] and the prediction of yield [2]. Therefore, the rapid and accurate prediction of AGB is critical to managing agricultural activities efficiently [3].

The conventional manual field measurement of AGB involves destructive, time-consuming, and laborious sampling [4]. Given these constraints, prompt and accurate monitoring of AGB is critical. Previous studies demonstrated that multispectral or hyperspectral information from satellites or airborne platforms has been widely used to monitor leaf area index [5], crop growth [6], nitrogen content [7], and the biomass of wheat [8]. However, unfavorable weather conditions, such as clouds or fog, may lead to a lack of appropriate satellite data, thereby limiting application in crop monitoring. In particular, high temporal resolution is required to explain spatial specificity in the field during the critical stage of crop monitoring [9]. The accuracy of phenological information needs to be determined from remote sensing observations, which depend largely on the frequency of

observations [2]. Moreover, data from remote sensing satellites are usually expensive and require extensive processing experience.

In recent years, with the development of unmanned aerial vehicles (UAVs) and their application in the field of remote sensing, the use of canopy spectra and UAV images has become a novel method for crop monitoring. For example, researchers have demonstrated the feasibility of using canopy spectra extracted from UAV hyperspectral images combined with partial least squares (PLS) regression to estimate the chlorophyll content of wheat [10]. In addition, color index and crop surface models have been extracted using orthogonal correction with (red–green–blue) RGB UAV images to estimate leaf area index (LAI) [11], plant height [12], and plant nitrogen content [13]. However, the spectral or image features obtained from a UAV image are saturated in the later stage of crop growth, leading to poor accuracy in the estimation of crop yield. To solve this problem, researchers have attempted to combine spectral and image features; it has been reported that vegetation indices (VIs) combined with a textural feature index (a normalized differential texture index, or NDTI) extracted from 550 and 800 nm band images obtained by a UAV multispectral camera provided better results than using traditional textural features and vegetation indices in a rice AGB estimation model [14]. Previous studies proved the feasibility of UAV-based textural features and VIs along with their combination in wheat AGB estimation and yield detection. However, crop canopy information cannot be fully obtained in the case of large crop coverage during the later stage of growth due to the low resolution of the images obtained by most UAVs, resulting in low accuracy in estimating wheat AGB at the later stage of growth [8]. The accurate estimation of crop AGB by using features obtained by a UAV platform needs to be improved at the later stage of crop growth.

Near infrared spectroscopy is one of the common methods used to detect crop biomass [15]. Using a hand-held spectrometer to obtain crop canopy reflectance and extracting a VI or effective wavelength to estimate wheat biomass has been proven to be an effective method [16]. For example, a power function, or exponential function relationship, was found between the specific vegetation index (RVI) and AGB of soybean at the seedling stage [17]. However, when crop biomass reaches a certain range, the crop canopy reflectance tends to be saturated, which leads to the low accuracy of estimated AGB in VI-based crops model. To reduce the effects of spectral saturation on crop AGB estimation, some researchers have used PLS regression based on band depth and VIs to estimate wheat biomass. However, the problem of canopy spectral saturation still exists [18]. Image technology based on consumer-grade digital cameras has been commonly used to monitor crop morphology [19], nutrient components [20], and pest status [21]. For example, image information can be obtained through a digital camera, and the image can be processed into 3D point cloud data to estimate wheat biomass, crown height, and harvest index. This measurement method has the advantages of high adaptability and robustness [22]. However, when plants grow to a certain level, the model used to estimate AGB based on image features performs poorly. The results demonstrate that an estimation model based on individual spectral or image features cannot accurately estimate crop AGB at the later stage of growth.

This study aims to evaluate the application of combining ground-scale images with spectral information in estimating the AGB of winter wheat. Image texture, spectral features, and their combinations are used to estimate the AGB of winter wheat in multiple growth stages. In this study, several methods that can be used to predict the AGB of winter wheat are proposed based on (1) spectral features (VIs, effective wavelength), (2) the optimal texture subset (OTEXS) and optimal normalized difference texture index subset (ONDTIS) calculated from canopy and side images, and (3) the combination of features with random forest (RF) regression and PLS regression.

## 2. Methods

### 2.1. Field Experiment and Measurements of Aboveground Biomass

The experiment was conducted at the Demonstration Base of National Precision Agriculture located in Xiaotangshan Town, Changping District (40°00′–40°21′ N, 116°34′–117°00′ E), Beijing, China (Figure 1). Ten wheat varieties, including Wanmai 38, Zhongmai 11, Lunxuan 518, and seven others, were adopted in the experiment that ran from October 2013 to June 2014. Each variety was transplanted at two planting densities, namely, 330 and 165 kg/hm². Forty plots, each with an area of 0.5 m × 0.6 m, were selected from forty sample plots as experimental objects. Five planting periods were selected for field experiments (seedling, jointing, heading, flowering, and filling stages).

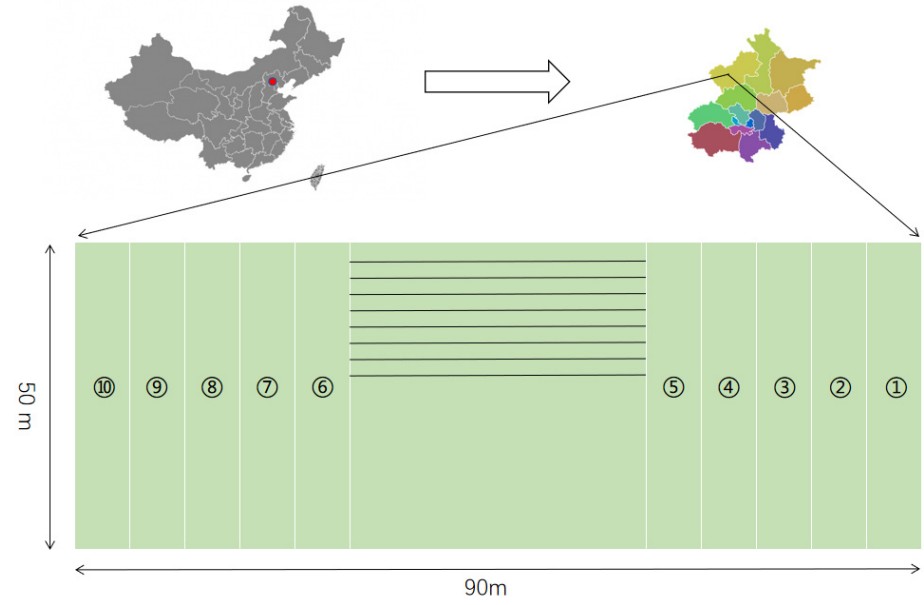

**Figure 1.** Experimental plots. Note: ①, Zhongmai 175; ②, Lunxuan 987; ③, Shixin 828; ④, Lunxuan 518; ⑤, Zhongmai 12; ⑥, Zhongmai 11; ⑦, Wanmai 38; ⑧, Zhongmai 13; ⑨, Jingdong 8; ⑩, Jimai 20. These are the varieties of wheat. The m stands for meter.

Ground destructive sampling was used to harvest wheat in each measurement area on the ground. The fresh weight of each sample was measured by an electronic scale with an accuracy of 0.5 g and a range of 5 kg. Altogether, 200 winter wheat AGB samples were obtained (40 samples in each of the five stages listed above). Table 1 shows the statistical data for measurements of winter wheat AGB.

**Table 1.** Descriptive statistics of aboveground biomass measurements at different growth stages.

| Dataset | Stage | Samples | Max/kg | Mean/kg | Min/kg | Std/kg |
|---|---|---|---|---|---|---|
| Calibration | Seedling | 27 | 0.23 | 0.102 | 0.04 | 0.05 |
| | Post-seedling | 107 | 0.73 | 0.37 | 0.08 | 0.13 |
| | All | 134 | 0.73 | 0.317 | 0.04 | 0.16 |
| Validation | Seedling | 13 | 0.232 | 0.106 | 0.047 | 0.054 |
| | Post-seedling | 53 | 0.641 | 0.372 | 0.121 | 0.125 |
| | All | 66 | 0.639 | 0.317 | 0.047 | 0.156 |

### 2.2. Digital Image and Spectrum Data Acquisition

Two different types of data were collected during two different time periods, as described below. The image data of each wheat canopy sample were acquired using a UC-M3151 industrial camera (MicroVision, Beijing, China) with an image resolution of 3 megapixels at the seedling stage. Photos were acquired on clear, cloudless, or partly

cloudy days; the specific time was 6:00–8:00 Beijing time. A white rectangular frame with an area of 0.5 m × 0.6 m and a tripod with adjustable height were built as auxiliary devices for field image acquisition to ensure the consistency of shooting angles and the height of multiple lenses. The tripod was adjusted to make the photos perpendicular to the wheat canopy and ensure they were taken 1 m from the canopy at the seedling stage. At the post-seedling stages (jointing to filling stages), the height of wheat plants increased continuously, the wheat leaves between different rows began to overlap with each other, and the visual field of the canopy image that was captured was almost filled with wheat leaves, leading to small visible changes in the proportion of wheat in the canopy in the background image. Therefore, winter wheat AGB was predicted by taking single-row side images of wheat and extracting textural features. The side image acquisition auxiliary device included a white background plate with an area of 1.2 m × 1.2 m and a tripod with adjustable height. When taking photographs of winter wheat, a white background plate was placed parallel to a row of wheat on the outside of the marking area and stood vertically on the ground (Figure 2). The tripod's height was adjusted to about 0.5 m, the lens was parallel to the white back plate, and the distance from the side of the wheat was about 1.2 m.

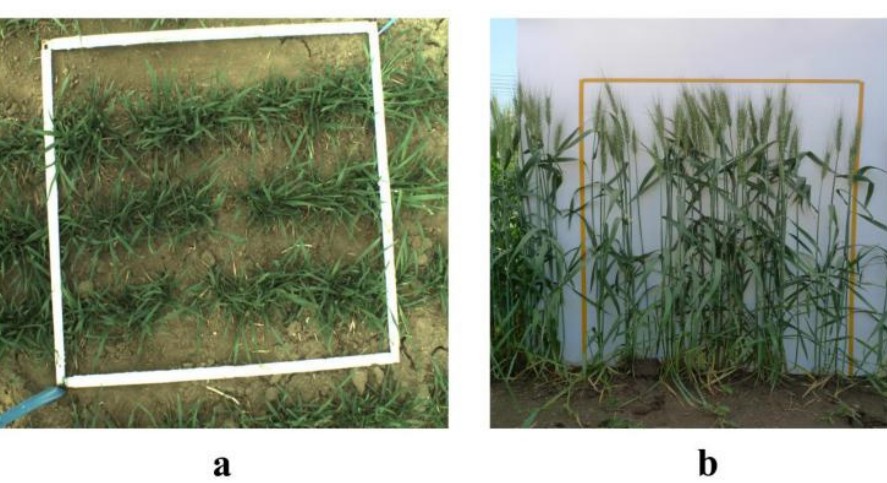

**Figure 2.** (**a**) Canopy image at the seedling stage; (**b**) side image at the post-seedling stage.

An AvaSpec-2048×14 fiber optic spectrometer (Avantes, Apeldoorn, The Netherlands) was used to collect spectral reflectance data with a wavelength range of 200–1100 nm and a spectral resolution of 2.4 nm. The spectral reflectance experiments were conducted on clear, cloudless, or partly cloudy days; the specific time was from 10:30 to 14:30 (Beijing time). When collecting the spectral data, the optical fiber probe was placed at about 0.75 m vertically below the wheat leaf canopy. Standard reference plate correction was performed every 15 min. Each plot was repeatedly measured five times, and the average value was regarded as the spectrum measurement result for each plot.

### 2.3. Data Processing

#### 2.3.1. Spectral Data Processing

Because the reflectance of the canopy experienced significant interference from the reflectance of the 200–400 nm and 1000–1100 nm bands, the canopy spectral reflectance of 400–1000 nm was selected for use in measuring the effective spectral parameters. Savitzky–Golay (S-G) smoothing can reduce the influence of high-frequency noise on the spectrum by averaging the multi-point spectral data; in addition, multiplicative scattering correction (MSC) can effectively eliminate the spectral difference caused by the different scattering levels of different spectral bands. A combination of MSC and S-G smoothing was used to preprocess the spectral data to reduce the influence of background noise and spectral scattering, thereby enhancing the correlation between the spectrum and the winter wheat AGB.

2.3.2. Vegetation Index Calculation

Six commonly used VIs (Table 2) were selected to evaluate the capability of spectral information to estimate AGB. The selected VIs were based on bands at 440, 680, 750, and 810 nm of canopy reflectance.

**Table 2.** Selected vegetation indices in this study for aboveground biomass estimation.

| Vegetation Index | Formula | Reference |
|---|---|---|
| OSAVI | $(1 + 0.16)(R810 - R750)/(R810 + R750 + 0.16)$ | [23] |
| NDVI | $(R810 - R680)/(R810 + R680)$ | [24] |
| DATT | $(R810 - R720)/(R810 - R680)$ | [25] |
| CI red edge | $(R810/R750) - 1$ | [26] |
| EVI | $2.5(R810 - R750)/(1 + R810 + 6R750 - 7.5R440)$ | [27] |
| RDVI | $(R810 - R750)/\sqrt{R810 + R750}$ | [28] |

Note: OSAVI, optimized soil-adjusted vegetation index; NDVI, normalized difference vegetation index; DATT, chlorophyll content index; CI red edge, red edge chlorophyll index; EVI, enhanced vegetation index; RDVI, renormalized difference vegetation index.

2.3.3. Texture Measurements

The gray level co-occurrence matrix (GLCM) can explore textural features and is often used to extract image textural features. Eight kinds of GLCM-based textures in three bands of each digital image, namely, red (R), green (G), and blue (B), were calculated at different stages by using ENVI software:

$$VAR = \sum_i \sum_j (i - u)^2 p(i,j) \tag{1}$$

$$HOM = i \sum_j \frac{1}{1 + (I - J)^2 p(i,j)} \tag{2}$$

$$CON = \sum_{n=0}^{N_g-1} n^2 \left\{ \sum_{i=1}^{N_g} \sum_{j=1}^{N_g} p(i,j), \ |i - j| = n \right\} \tag{3}$$

$$ENT = -\sum_i \sum_j p(i,j) \log(p(i,j)) \tag{4}$$

$$SEM = \sum_i \sum_j \{p(i,j)\}^2 \tag{5}$$

$$COR = \frac{\sum_i \sum_j (i,j) p(i,j) - u_x u_y}{\sigma_x \sigma_y} \tag{6}$$

$$MEAN = \frac{\sum_i \sum_j p(i,j)}{ij}, \tag{7}$$

$$DIS = \sum_{n=1}^{N_g-1} n \left\{ \sum_{i=1}^{N_g} \sum_{j=1}^{N_g} p(i,j), \ |i - j| = n \right\} \tag{8}$$

where p(i,j): (i,j)th entry in a normalized gray-tone spatial-dependence matrix = P(i,j)/R. $N_g$ represents the number of distinct gray levels in the quantized image. $U_x$, $U_y$, $\sigma_x$, and $\sigma_y$ are the means and standard deviations. Further details of the calculation are in available in [29].

In this paper, a minimum window size of $3 \times 3$ pixels was used for textural feature extraction to reduce computational complexity. In addition, the NDTI was used to explore its ability to predict AGB in winter wheat [7] and is defined as in Equation (9):

$$NDTI = (Ti - Tj)/(Ti + Tj) \tag{9}$$

where Ti and Tj represent two different randomly selected textures.

### 2.4. Regression Modelling Methods

2.4.1. Simple Linear Regression (LR)

Linear regression (LR) has a simple and easily understandable model structure, which has great applicability in practical applications [30]. AGB estimation of winter wheat based on an LR model was used to evaluate the prediction performance of EWs, VIs, texture, and NDTI in estimating AGB. The LR models were defined as shown in Equation (10):

$$AGB = bX + a \tag{10}$$

where X represents the single input predictor variable and b and a represent the slope and intercept of the fitted line of the LR model, respectively.

2.4.2. PLS and Random Forest (RF)

Partial least squares regression is a dimension reduction method that is a popular modelling approach frequently used in studies of vegetation because it provides an efficient way to make full use of spectral information [31,32]. Random forest (RF) is a powerful machine learning algorithm. Each tree in the RF is constructed using a deterministic algorithm by selecting a group of random variables and a random sample from a calibration dataset. RF can not only deal with a large number of input variables, but also uses a small number of variables to obtain a reasonable amount of prediction accuracy. In addition, RF regression is beneficial in overcoming the over-fitting problem of simple decision trees [33].

### 2.5. Statistical Analysis

This study analyzed the relationship between AGB and different types of variables (effective wavelength, VIs, OTEXS, ONDTIS, and a combination of spectral indices and image indices) to improve the applicability of biomass estimation models. The flowchart in Figure 3 illustrates the experimental method. A total of 200 experimental data points were collected and randomly divided into a training set and a test set, with 2/3 of the samples selected as a training dataset and 1/3 of the samples selected as a test dataset. Three regression models (LR, PLS, and RF) were established to estimate winter wheat biomass. The LR model was used to analyze the correlation between each calculated index and the AGB of winter wheat. The combined features were used as the input of the PLS and RF regression models to estimate the AGB of winter wheat. The AGB estimation performance of the three regression models was evaluated by the determination coefficient [$R^2$, Equation (11)] and the root mean square error [RMSE, Equation (12)]. Python software was used for all statistical analyses.

$$R^2 = 1 - \frac{\sum_{i=1}^{n}(y_i - \hat{y}_i)^2}{\sum_{i=1}^{n}(y_i - \overline{y}_i)^2} \tag{11}$$

$$RMSE = \sqrt{\frac{1}{n}\sum_{i=1}^{n}(y_i - \hat{y}_i)^2} \tag{12}$$

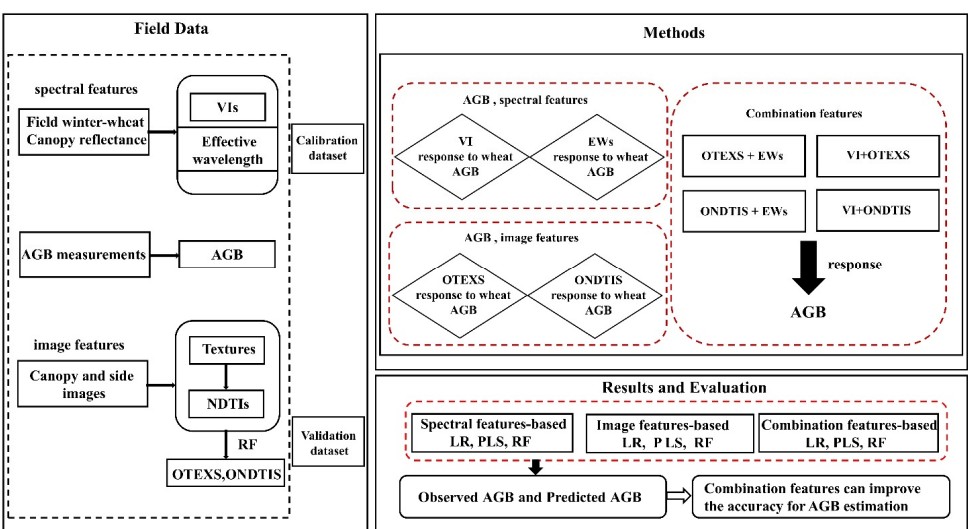

**Figure 3.** Experiment methodology.

## 3. Results

### 3.1. Estimation of AGB with Effective Wavelengths

The near infrared (400–1000 nm) reflectance spectra of the winter wheat canopy were recorded. S-G and MSC were selected to preprocess the spectrum and eliminate the random error and scattering effect. The interference of the treated spectral curve was obviously reduced (Figure 4). Meanwhile, the adjacent wavelengths were usually highly correlated. A successive projection algorithm (SPA) was used to select the effective wavelengths of the processed spectrum to reduce information redundancy and improve the utilization efficiency of spectral information.

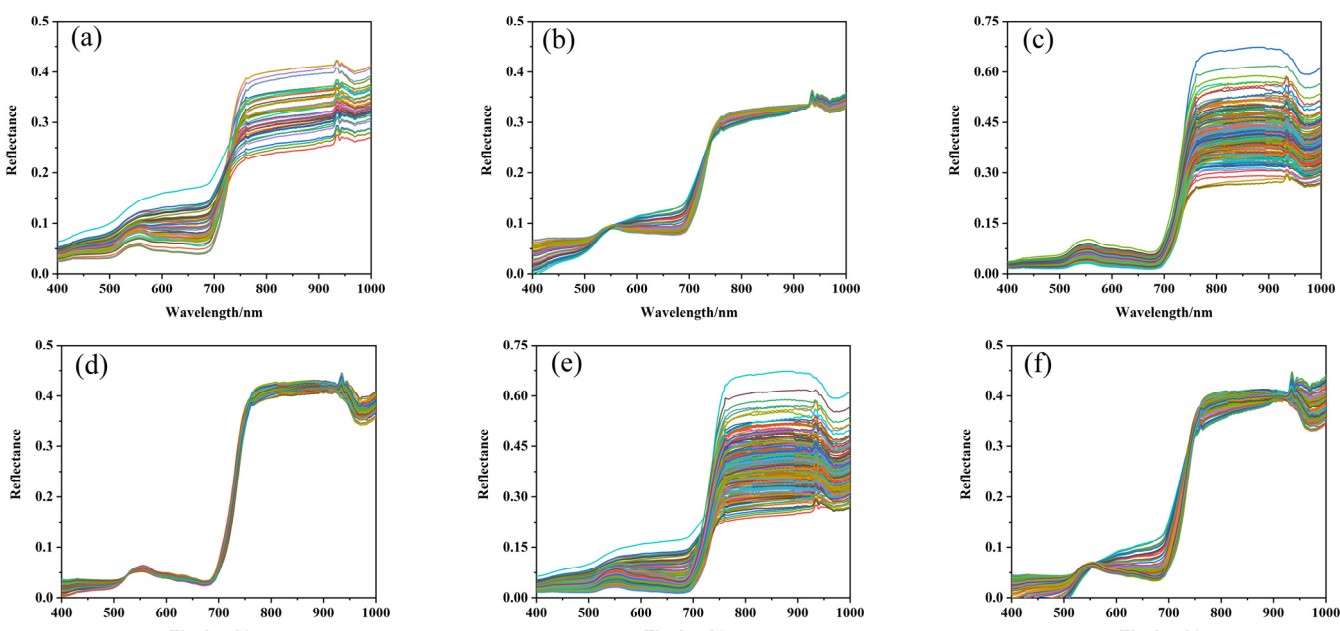

**Figure 4.** Reflectance spectra of the winter wheat canopy at different growth stages. Omni-spectral data in (**a**) the seedling stage, (**b**) the post-seedling stage, and (**c**) all stages, as well as pre-spectrum data in (**d**) the seedling stage, (**e**) post-seedling stage, and (**f**) all stages.

Figure 5 presents the results of effective wavelength selection using an SPA model at the seedling stage. The RMSE values of different subsets in the SPA model during the seedling stage are shown in Figure 5a, where "□" represents the number of the effective

wavelength. The results show that when the number of variables was less than four, the RMSE value shows a downward trend. In contrast, the change tends to be flat. Therefore, four effective wavelengths were extracted by the SPA at the seedling stage to estimate the AGB of winter wheat. Figure 5b shows the specific selection of effective wavelengths at the seedling stage, in which "□" represents the selected effective wavelengths (583.35 nm, 762.553 nm, 929.176 nm, and 940.639 nm). Similarly, the number of effective wavelengths selected by the SPA at the post-seedling stage and for all stages was 12 and 14, respectively.

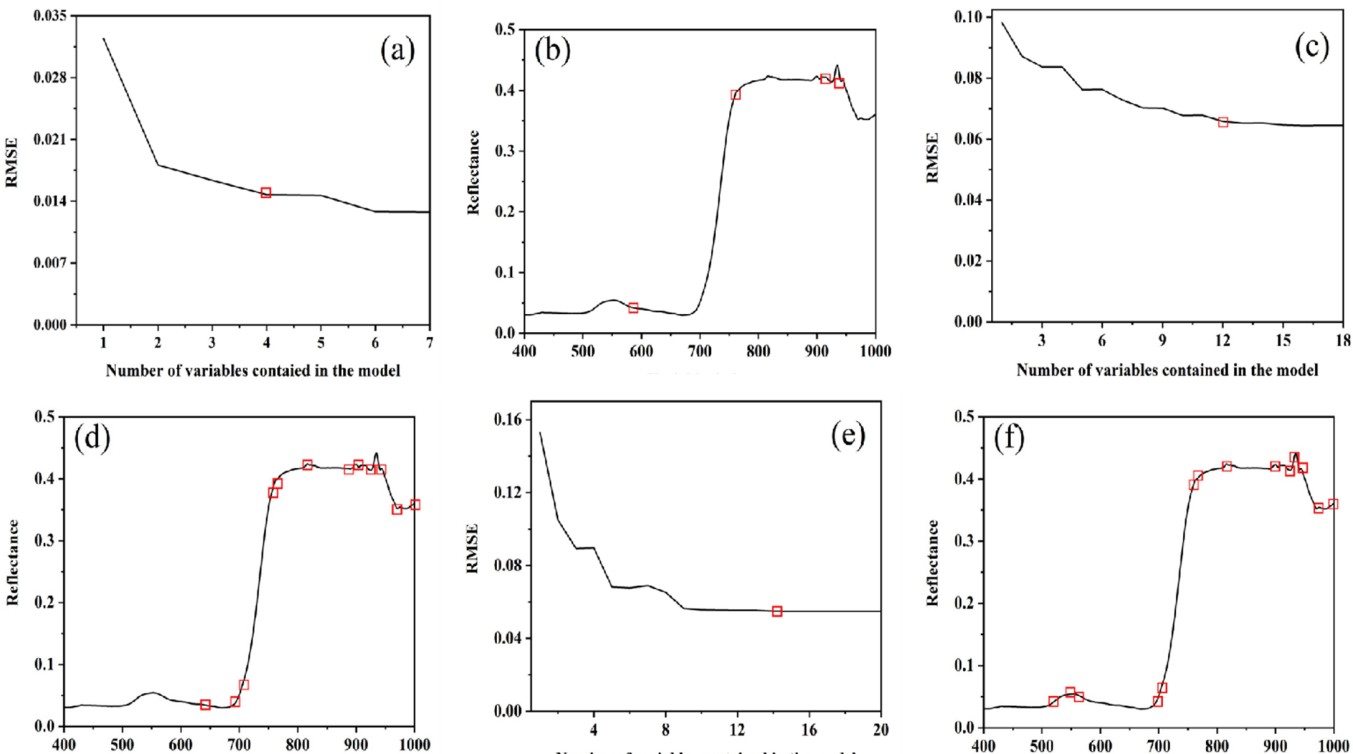

**Figure 5.** Reflectance spectra of the winter wheat canopy at the seedling stage: (**a**–**c**) show the number of variables contained in the model in the seedling stage, post-seedling stage, and at all stages, respectively; (**d**–**f**) show the wavelength in the seedling stage, post-seedling stage, and at all stages, respectively. A square (□) represents the number of the effective wavelength.

The results of the model that predicts the AGB of winter wheat based on the effective wavelengths (Table 3) at multiple growth stages. The results show that the LR model based on effective wavelength estimation of winter wheat AGB had a higher $R^2$ and lower RMSE value at the seedling stage ($R^2$ = 0.89, RMSE = 0.016), the post-seedling stage ($R^2$ = 0.73, RMSE = 0.07), and in all stages ($R^2$ = 0.83, RMSE = 0.06) (Figure 6).

**Table 3.** Aboveground biomass estimates using selected effective wavelengths at different growth stages.

| | LR | | | | RF | | | | PLS | | | |
|---|---|---|---|---|---|---|---|---|---|---|---|---|
| | $R_c^2$ | $RMSE_c$ | $R_v^2$ | $RMSE_v$ | $R_c^2$ | $RMSE_c$ | $R_v^2$ | $RMSE_v$ | $R_c^2$ | $RMSE_c$ | $R_v^2$ | $RMSE_v$ |
| Seedling | 0.89 | 0.01 | 0.89 | 0.01 | 0.96 | 0.01 | 0.84 | 0.02 | 0.91 | 0.01 | 0.89 | 0.01 |
| Post-seedling | 0.72 | 0.06 | 0.73 | 0.07 | 0.92 | 0.03 | 0.69 | 0.06 | 0.74 | 0.05 | 0.67 | 0.05 |
| All | 0.84 | 0.06 | 0.83 | 0.06 | 0.95 | 0.03 | 0.72 | 0.07 | 0.86 | 0.05 | 0.83 | 0.06 |

Note: $R_c^2$, the $R^2$ value of the calibration set; $R_v^2$, the $R^2$ value of the validation set; $RMSE_c$, the RMSE of the calibration set; $RMSE_v$, the RMSE of the validation set.

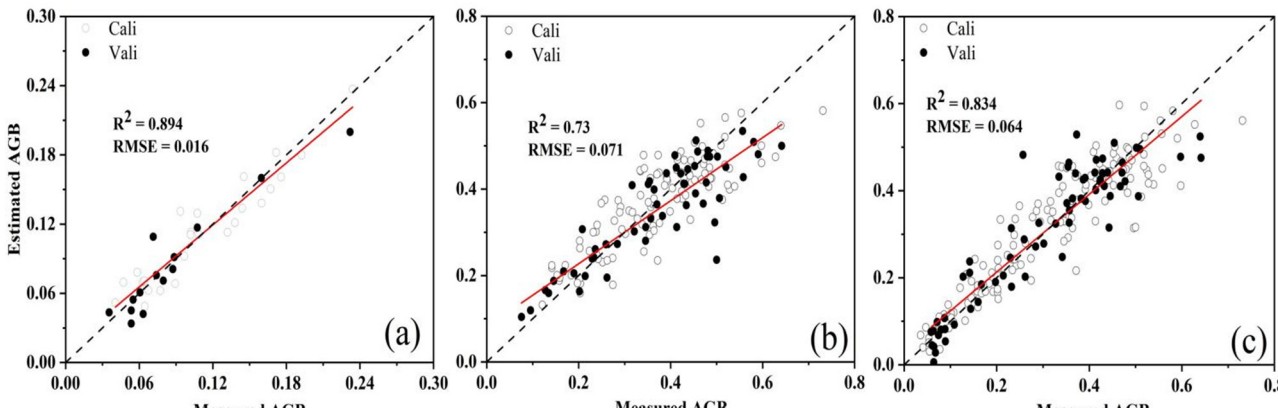

**Figure 6.** Validation of aboveground biomass (AGB) estimation models established by best performing effective wavelengths vs. AGB for the following stages: (**a**) seedling, (**b**) post-seedling, (**c**) all stage.

### 3.2. Estimation of AGB with VIs

Figure 7 shows the significant positive correlation between VI and winter wheat AGB. The AGB of winter wheat was predicted well by VIs with the LR, RF, and PLS models at the seedling stage. When all stages were considered, the prediction accuracy of winter wheat AGB at the post-seedling stage was lower (Table 4). The LR prediction model based on the renormalized difference vegetation index had the best prediction ability when estimating the AGB of winter wheat at the seedling stage ($R^2 = 0.82$, RMSE = 0.02). In all stages, the estimation of AGB by the LR model based on the optimized soil-adjusted vegetation index had a good prediction effect ($R^2 = 0.68$, RMSE = 0.09). However, with the continuous growth of winter wheat, the correlation between each individual VI and winter wheat AGB was low at the post-seedling stage (Figure 8). The accuracy of the predictive model of winter wheat AGB based on each individual vegetation index separately performed poorly, of which the LR model based on the enhanced vegetation index had the best predictability ($R^2 = 0.45$, RMSE = 0.1). The LR model based on a single vegetation index did not accurately estimate the AGB of winter wheat at the post-seedling stage.

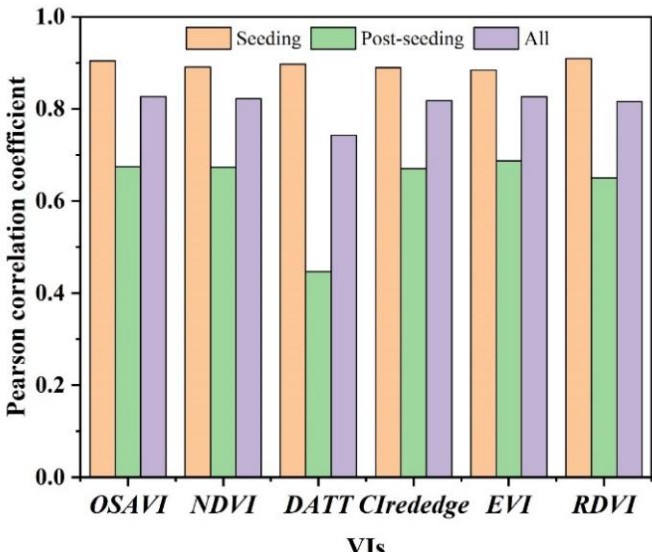

**Figure 7.** Pearson correlation coefficient between vegetation indices and aboveground biomass.

**Table 4.** Prediction results of LR, RF, and PLS with the optimal VI across all different growth stages.

| Stage | VI | LR | | | | RF | | | | PLS | | | |
|---|---|---|---|---|---|---|---|---|---|---|---|---|---|
| | | $R^2_c$ | $RMSE_c$ | $R^2_v$ | $RMSE_v$ | $R^2_c$ | $RMSE_c$ | $R^2_v$ | $RMSE_v$ | $R^2_c$ | $RMSE_c$ | $R^2_v$ | $RMSE_v$ |
| Seedling | RDVI | 0.83 | 0.02 | 0.8 | 0.02 | 0.94 | 0.01 | 0.82 | 0.02 | 0.9 | 0.01 | 0.82 | 0.02 |
| Post-Seedling | EVI | 0.45 | 0.08 | 0.45 | 0.1 | 0.89 | 0.04 | 0.4 | 0.09 | 0.48 | 0.06 | 0.43 | 0.05 |
| All | OSAVI | 0.68 | 0.08 | 0.67 | 0.08 | 0.93 | 0.03 | 0.68 | 0.09 | 0.7 | 0.07 | 0.63 | 0.07 |

Note: $R^2_c$, the $R^2$ value of the calibration set; $R^2_v$, the $R^2$ value of the validation set; $RMSE_c$, the RMSE of the calibration set; $RMSE_v$, the RMSE of the validation set.

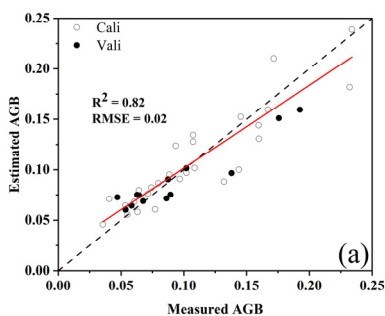 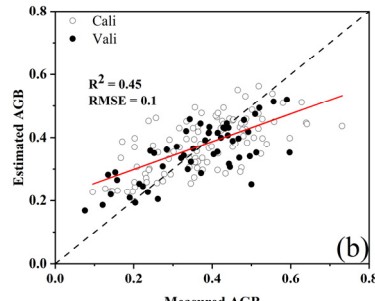 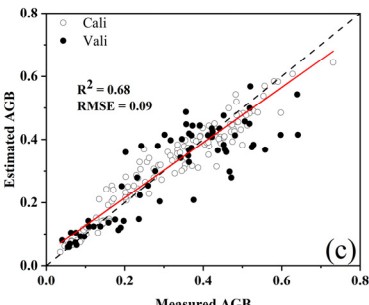

**Figure 8.** Validation of aboveground biomass (AGB) estimation models established by best performing vegetation index vs. AGB for the following stages: (**a**) seedling, (**b**) post-seedling, (**c**) all stages.

### 3.3. Estimation of AGB with Textural Features

The relationship between the textural features of the R, G, and B bands along with the AGB of winter wheat varied greatly at multiple growth stages (Table 5). Most single textures had poor prediction performance for estimating AGB at all stages. At the seedling stage, the textural features of the R, G, and B bands (mean, variation, Con, Dis) were significantly correlated with AGB. At the post-sowing stage, the correlation between the textural features of the three bands (Hom, Ent, and Sem) and winter wheat AGB decreased significantly. Therefore, a single textural feature cannot predict AGB comprehensively and accurately, which is consistent with using a single vegetation index to predict the AGB of winter wheat.

**Table 5.** Relationships between aboveground biomass and grey level co-occurrence matrix-based texture measurements with the calibration set ($R^2$).

| | Red Band | | | Green Band | | | Blue Band | | |
|---|---|---|---|---|---|---|---|---|---|
| | Seedling | Post-Seedling | All | Seedling | Post-Seedling | All | Seedling | Post-Seedling | All |
| Mean | 0.736 | 0.392 | 0.384 | 0.715 | 0.326 | 0.24 | 0.725 | 0.509 | 0.02 |
| Var | 0.787 | 0.017 | 0.004 | 0.685 | 0.033 | 0.004 | 0.603 | 0.022 | 0.095 |
| Hom | 0.567 | 0.562 | 0.074 | 0.559 | 0.582 | 0.001 | 0.56 | 0.599 | 0.262 |
| Con | 0.787 | 0.014 | 0.017 | 0.674 | 0.026 | 0.008 | 0.6 | 0.02 | 0.042 |
| Dis | 0.791 | 0.253 | 0.001 | 0.717 | 0.308 | 0.009 | 0.715 | 0.314 | 0.169 |
| Ent | 0.236 | 0.572 | 0.013 | 0.233 | 0.59 | 0.001 | 0.361 | 0.602 | 0.091 |
| Sem | 0.04 | 0.555 | 0.017 | 0.033 | 0.577 | 0.001 | 0.152 | 0.575 | 0.076 |
| Cor | 0.005 | 0.006 | 0.004 | 0.001 | 0.004 | 0.191 | 0.019 | 0.004 | 0.017 |

The LR model based on a single textural feature performed well at the seedling stage. The correlation between most textural features and winter wheat AGB was not significant during the post-seedling stage or for all combined stages. Therefore, this paper proposes the use of the OTEXS as the input for the winter wheat AGB prediction model to improve the ability of the LR model to estimate wheat AGB based on a single textural feature. We employed the RF model to calculate texture feature importance, as well as to sort and select the best textural features to form the optimal textural feature subset. The number of OTEXSs at different growth stages was six.

When compared with the prediction model of winter wheat AGB based on texture features, the estimation accuracy of the LR, RF, and PLS prediction model based on the OTEXS was significantly improved at the post-seedling stage and for all stages (Table 6). The accuracy of the OTEXS-based RF model was significantly improved at the post-seedling stage ($R^2 = 0.73$, RMSE = 0.07) and for all stages ($R^2 = 0.75$, RMSE = 0.06). The LR model of AGB based on the OTEXS also provided the greatest estimation performance at the seedling stage ($R^2 = 0.85$, RMSE = 0.01). The results show that the OTEXS is one of the more promising schemes that can be used to improve the prediction accuracy of an LR model for estimating AGB at multiple growth stages (Figure 9).

**Table 6.** Prediction results of LR, RF, and PLS with the optimal texture subset across all different growth stages.

| | LR | | | | RF | | | | PLS | | | |
|---|---|---|---|---|---|---|---|---|---|---|---|---|
| | $R_c^2$ | $RMSE_c$ | $R_v^2$ | $RMSE_v$ | $R_c^2$ | $RMSE_c$ | $R_v^2$ | $RMSE_v$ | $R_c^2$ | $RMSE_c$ | $R_v^2$ | $RMSE_v$ |
| Seedling | 0.90 | 0.01 | 0.85 | 0.01 | 0.96 | 0.01 | 0.83 | 0.02 | 0.91 | 0.01 | 0.85 | 0.01 |
| Post-seedling | 0.76 | 0.06 | 0.66 | 0.07 | 0.93 | 0.03 | 0.73 | 0.07 | 0.77 | 0.05 | 0.65 | 0.06 |
| All | 0.81 | 0.06 | 0.76 | 0.06 | 0.94 | 0.03 | 0.75 | 0.06 | 0.82 | 0.05 | 0.77 | 0.07 |

Note: $R_c^2$, the $R^2$ value of the calibration set; $R_v^2$, the $R^2$ value of the validation set; $RMSE_c$, the RMSE of the calibration set; $RMSE_v$, the RMSE of the validation set.

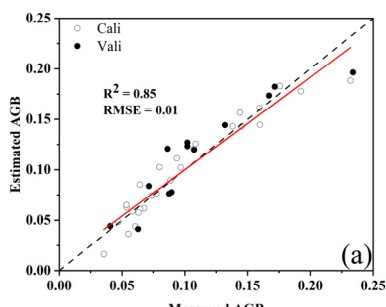 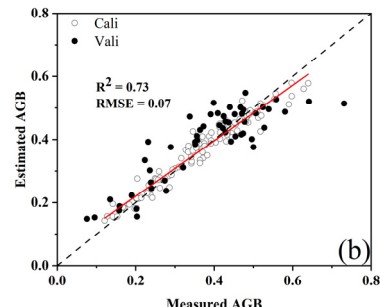 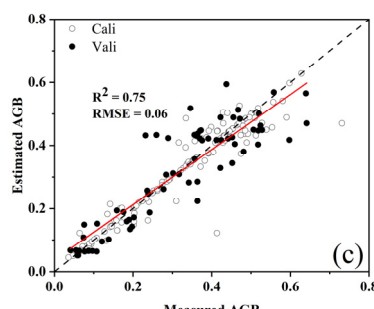

**Figure 9.** Validation of aboveground biomass (AGB) estimation models established by best performing optimal texture subset vs. AGB for the following stages: (**a**) seedling, (**b**) post-seedling, (**c**) all stages.

In this study, the researchers attempted to combine two different textures randomly to construct a new texture parameter NDTI for AGB estimation [7]. Figure 10 shows that the $R^2$ value of the LR models based on the NDTI for estimating AGB had significant differences at multiple growth stages. When compared with the two other growth stages, the LR model based on the NDTI performed best for winter wheat AGB estimation at the seedling stage. The NDTI (Rmean, Gmean) had the best performance, with an $R^2$ value of 0.94. At the post-seedling stage and across all stages, the NDTI-based LR model also achieved better performance in the estimation of AGB than a single textural feature. Hence, the NDTI (Bean, Bhom) performed best, with an $R^2$ value of 0.67 at the post-seedling stage; moreover, the NDTI (Ghom, Bhom) also performed best across all stages, with an $R^2$ value of 0.51. Table 7 shows the five best performing NDTIs at multiple growth stages.

The low accuracy of the AGB prediction model based on a single NDTI was detected at the post-seedling stage and across all stages. This study employed the RF model to calculate feature importance, as well as to sort and select the NDTI with the highest importance as the optimal NDTI subset (ONDTIS) for estimating AGB. The number of NDTIs of the ONDTIS in each growth stage was 8, 13, and 12, respectively. Compared with an individual NDTI, the PLS models of AGB based on the ONDTIS were improved at multiple stages (Table 8). At the seedling stage, the ONDTIS performed best, with the highest $R^2$ value of 0.91 and the lowest RMSE of 0.01. Excellent results were obtained at the post-seedling stage and across all stages (Figure 11).

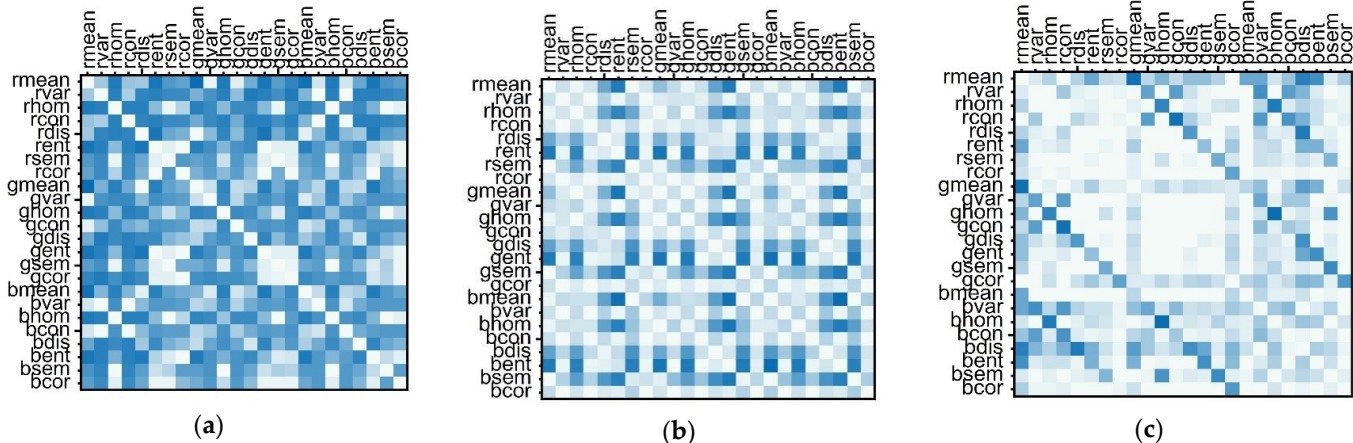

**Figure 10.** $R^2$ value of the normalized differential texture index-based linear regression models for estimating aboveground biomass at the (**a**) seedling stage, (**b**) post-seedling stage, and (**c**) across all stages.

**Table 7.** Performance of the five best normalized differential texture index-based linear regression models based on the calibration dataset.

| | Seedling | | | Post-Seedling | | | All | |
|---|---|---|---|---|---|---|---|---|
| x1 | x2 | $R^2$ | x1 | x2 | $R^2$ | x1 | x2 | $R^2$ |
| Rmean | Rcor | 0.93 | Bmean | Bent | 0.67 | Ghom | Bhom | 0.51 |
| Rmean | Gcor | 0.9 | Gent | Bmean | 0.64 | Rmean | Rcor | 0.48 |
| Rmean | Bdis | 0.86 | Rent | Gcor | 0.64 | Rcon | Gcon | 0.48 |
| Rmean | Gdis | 0.85 | Gmean | Bent | 0.62 | Rhom | Bhom | 0.45 |
| Rdis | Gent | 0.84 | Gmean | Gent | 0.62 | Rdis | Bdis | 0.45 |

Note: x1 and x2 represent two different randomly selected textures.

**Table 8.** Prediction results of LR, RF, and PLS with the optimal normalized difference texture index subset across all different growth stages.

| | LR | | | | RF | | | | PLS | | | |
|---|---|---|---|---|---|---|---|---|---|---|---|---|
| | $R_c^2$ | $RMSE_c$ | $R_v^2$ | $RMSE_v$ | $R_c^2$ | $RMSE_c$ | $R_v^2$ | $RMSE_v$ | $R_c^2$ | $RMSE_c$ | $R_v^2$ | $RMSE_v$ |
| Seedling | 0.95 | 0.01 | 0.86 | 0.019 | 0.95 | 0.01 | 0.82 | 0.02 | 0.93 | 0.01 | 0.91 | 0.01 |
| Post-seedling | 0.82 | 0.05 | 0.74 | 0.07 | 0.94 | 0.02 | 0.73 | 0.06 | 0.78 | 0.05 | 0.75 | 0.04 |
| All | 0.82 | 0.066 | 0.77 | 0.07 | 0.94 | 0.03 | 0.82 | 0.06 | 0.82 | 0.05 | 0.78 | 0.06 |

Note: $R_c^2$, the $R^2$ value of the calibration set; $R_v^2$, the $R^2$ value of the validation set; $RMSE_c$, the RMSE of the calibration set; $RMSE_v$, the RMSE of the validation set.

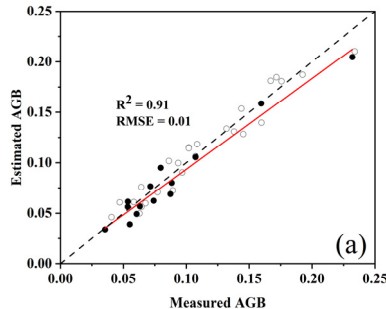
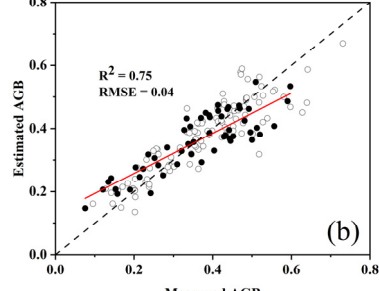
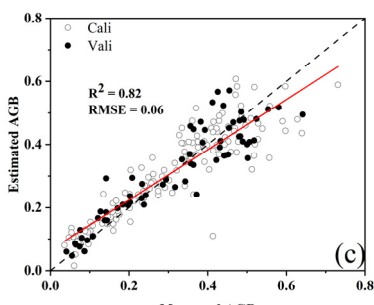

**Figure 11.** Validation of aboveground biomass (AGB) estimation models established by best performing optimal normalized difference texture index subset vs. AGB for the following stages: (**a**) seedling, (**b**) post-seedling, (**c**) all stages.

### 3.4. Estimation of Winter Wheat AGB with Combination Features

The winter wheat AGB prediction models based on a single spectral or image feature at the post-seedling stage and across all stages had limitations and low accuracy. A multiple regression prediction model of winter wheat AGB was established by combining spectral and image information to provide appropriate technical support so as to improve the estimation accuracy of winter wheat AGB at the post-seedling stage. The combination feature was formed by the combination of effective wavelengths and VIs, the OTEXS, and the ONDTIS. The results of estimating winter wheat AGB based on the combined features at multiple growth stages are shown in Table 9. The PLS regression estimation model of AGB based on the combination of the OTEXS and effective wavelengths had the highest accuracy at the seedling stage ($R^2$ = 0.943, RMSE = 0.01). The LR estimation model of AGB based on the combination of the OTEXS and effective wavelengths had the highest accuracy at the post-seedling stage ($R^2$ = 0.78, RMSE = 0.06; Figure 12). The RF-based winter wheat AGB prediction model based on the combination of the OTEXS and effective wavelengths also had the highest accuracy across all stages ($R^2$ = 0.87, RMSE = 0.05). Figure 13 reveals the accuracy of the assessment results of the multivariate regression models for AGB estimation based on the independent validation datasets. As shown in Figure 13a–i, for the three regression methods (LR, RF, and PLS), the accuracy of the models based on combined features was better than that of models based on image textures or spectral features in estimating winter wheat AGB, especially at the post-seedling stage.

**Table 9.** Prediction results of LR, RF, and PLS with combined features across all different growth stages.

| Stage | Features | LR | | | | PLS | | | | RF | | | |
|---|---|---|---|---|---|---|---|---|---|---|---|---|---|
| | | $R_c^2$ | $RMSE_c$ | $R_v^2$ | $RMSE_v$ | $R_c^2$ | $RMSE_c$ | $R_v^2$ | $RMSE_v$ | $R_c^2$ | $RMSE_c$ | $R_v^2$ | $RMSE_v$ |
| Seedling | VI + OTEXS | 0.95 | 0.01 | 0.85 | 0.02 | 0.93 | 0.01 | 0.92 | 0.01 | 0.97 | 0.007 | 0.88 | 0.02 |
| | EWs + OTEXS | 0.98 | 0.01 | 0.85 | 0.02 | 0.96 | 0.009 | 0.94 | 0.01 | 0.97 | 0.008 | 0.82 | 0.02 |
| | ONDTIS + EWs | 0.96 | 0.01 | 0.94 | 0.01 | 0.97 | 0.007 | 0.89 | 0.01 | 0.97 | 0.007 | 0.77 | 0.02 |
| | ONDTIS + VI | 0.95 | 0.01 | 0.87 | 0.01 | 0.93 | 0.01 | 0.91 | 0.01 | 0.97 | 0.007 | 0.77 | 0.02 |
| Post-seedling | VI + OTEXS | 0.77 | 0.06 | 0.72 | 0.06 | 0.80 | 0.05 | 0.71 | 0.06 | 0.95 | 0.02 | 0.70 | 0.06 |
| | EWs + OTEXS | 0.82 | 0.05 | 0.78 | 0.05 | 0.84 | 0.04 | 0.76 | 0.05 | 0.96 | 0.02 | 0.74 | 0.06 |
| | ONDTIS + EWs | 0.82 | 0.05 | 0.8 | 0.05 | 0.83 | 0.04 | 0.71 | 0.06 | 0.96 | 0.02 | 0.77 | 0.05 |
| | ONDTIS + VI | 0.8 | 0.05 | 0.74 | 0.06 | 0.80 | 0.05 | 0.74 | 0.05 | 0.95 | 0.02 | 0.72 | 0.06 |
| ALL | VI + OTEXS | 0.85 | 0.06 | 0.81 | 0.05 | 0.85 | 0.05 | 0.83 | 0.06 | 0.97 | 0.02 | 0.86 | 0.05 |
| | EWs + OTEXS | 0.88 | 0.05 | 0.85 | 0.05 | 0.89 | 0.04 | 0.83 | 0.06 | 0.97 | 0.02 | 0.87 | 0.05 |
| | ONDTIS + EWs | 0.88 | 0.05 | 0.87 | 0.05 | 0.89 | 0.04 | 0.82 | 0.06 | 0.97 | 0.02 | 0.85 | 0.05 |
| | ONDTIS + VI | 0.87 | 0.05 | 0.77 | 0.06 | 0.85 | 0.05 | 0.83 | 0.05 | 0.97 | 0.02 | 0.85 | 0.05 |

Note: $R_c^2$, the $R^2$ value of the calibration set; $R_v^2$, the $R^2$ value of the validation set; $RMSE_c$, the RMSE of the calibration set; $RMSE_v$, the RMSE of the validation set; VI, vegetation index; OTEXS, optimal texture subset; ONDTIS, optimal normalized difference texture index subset; EWs, effective wavelengths.

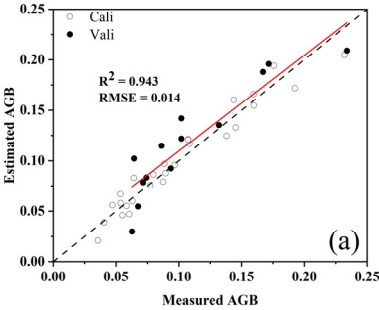 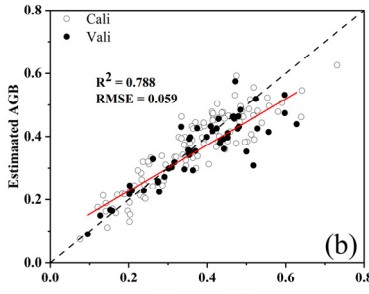 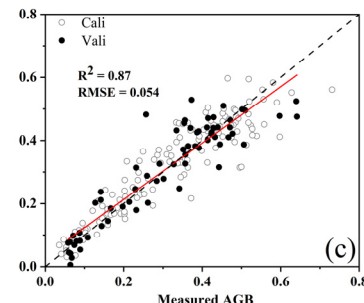

**Figure 12.** Validation of aboveground biomass (AGB) estimation models established by best performing optimal normalized difference texture index subset vs. AGB for the following stages: (**a**) seedling, (**b**) post-seedling, (**c**) all stages.

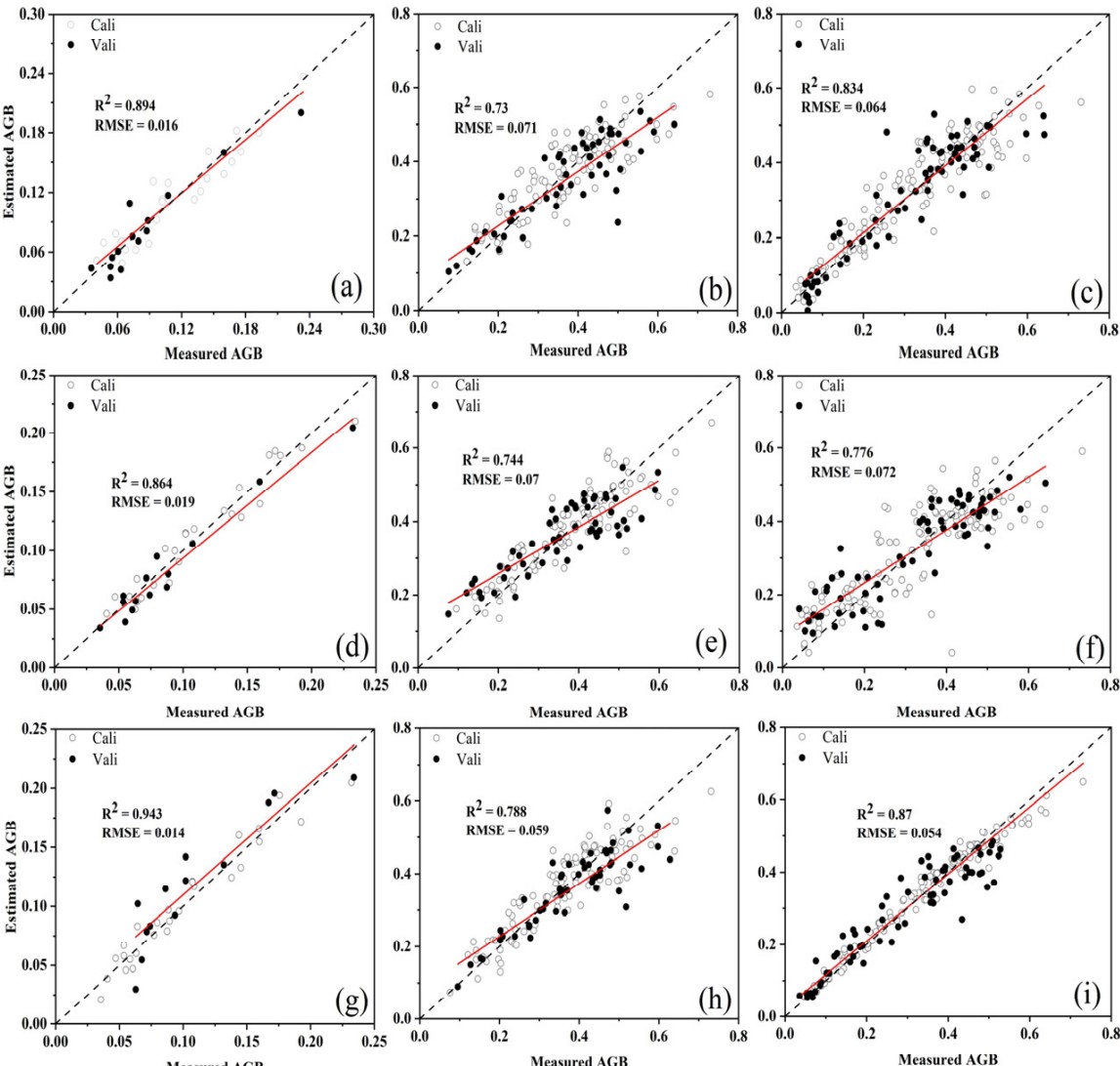

**Figure 13.** Validation of aboveground biomass (AGB) estimation models established by best performing spectral, image, and combination features. Effective wavelengths for the following stages: (**a**) seedling, (**b**) post-seedling, (**c**) all stages. ONDTIS for the following stages: (**d**) seedling, (**e**) post-seedling, (**f**) all stages. OTEX + effective wavelengths for the following stages: (**g**) seedling, (**h**) post-seedling, (**i**) all stages.

## 4. Discussion

### 4.1. Estimation of AGB with Spectral and Texture Features

Spectral techniques have been widely used to predict crop biomass and yield [30]. The prediction accuracy of the LR model based on effective wavelengths at the post-seedling stage ($R^2 = 0.73$, RMSE = 0.07) was improved when compared with results based on a vegetation index alone ($R^2 = 0.45$, RMSE = 0.1). The wavelength selected by the SPA includes red (550–770 nm) and near-infrared regions (800–1000 nm), which are the main absorption bands of plant photosynthesis after sowing as seedlings mature; however, VI only uses the spectral information of several bands. When compared with the AGB model at the seedling stage ($R^2 = 0.89$, RMSE = 0.01), the accuracy of the AGB prediction model had poor performance ($R^2 = 0.73$, RMSE = 0.07) at the post-seedling stage, similar to the model based on VIs. As the canopy matures, the increase in the leaf layer leads to an increase in canopy complexity such that shadow becomes a spectral trap of incident energy and reduces the amount of radiation returned to the sensor [34]. AGB cannot be comprehensively and accurately predicted based on a single spectral feature.

In recent years, researchers have found that image texture features can improve the effect of using satellite images to predict forest biomass [35]. In the present work, the correlation between a single textural feature and AGB was quite different at multiple stages, which is consistent with a previous report [36]. The AGB prediction model based on a single textural feature (mean and dis) at the seedling stage resulted in the best $R^2$ value (Table 5). The canopy of winter wheat had rich structural information at the seedling stage. The mean and dis values contain the average value in the moving window of the target and the background, which can smooth an image and minimize background interference [7]. In this article, most of the textural features have poor correlations with the AGB of winter wheat. This result provided a similar conclusion to that found in the correlation between textural features and AGB in temperate forests [37]. By aiming at the poor correlation between textural features and AGB, the NDTI-based rice AGB prediction model had great performance at multiple stages [7]. A total of 276 NDTIs were calculated. As a result, the correlation between NDTIs and AGB was found to be different at multiple stages. Compared with textural features, the correlation between most NDTIs and AGB was obviously improved in multiple stages, while the accuracy of the AGB prediction model based on these NDTIs was better than that of the AGB model based on textural features. The AGB prediction model based on NDTIs (Rmean and Rcon) at the seedling stage had the optimal effect ($R^2$ = 0.93). The correlation between NDTIs and AGB at the seedling stage was better than that at the late stage and across all stages, which is consistent with the results of the analysis of textural features. The canopy structure information was most accurately captured by digital images at the seedling stage, in which the density of plants was low and adjacent leaves hardly overlapped. However, with the continuous growth of winter wheat, the canopy leaves of each plant overlapped with one another, resulting in a loss of structural information in digital images and the decreased accuracy of the winter wheat AGB prediction model based on image features. The result is consistent with previous studies that used convolutional neural networks to predict biomass-related shapes [8]. Hence, AGB cannot be predicted comprehensively and accurately by using a single image feature.

Compared with the models based on a single spectral or image feature, the AGB model based on the effective wavelength of a single spectral feature was better than the model based on a single image feature at the seedling stage and across all stages. However, at the post-seedling stage, the accuracy of the AGB model based on the ONDTIS was better than the spectral feature model. Ref. [38] found that a large amount of spectral information is lost at the later stages of crop growth, leading to difficulty in accurately predicting the yield. Therefore, image and spectral information sources have different effects on the AGB prediction model at multiple growth stages. At the seedling stage, the spectral and image information has a strong correlation with AGB due to the simple three-dimensional structure of the crop canopy. By contrast, the correlation between the spectral reflectance of the canopy and the AGB of winter wheat decreased because the leaves obstructed one another, leading to a loss of canopy information during the growth of the crop canopy.

### 4.2. Estimation of AGB with Optimal Image Features

In view of the poor prediction performance of the AGB model based on a single textural feature and single NDTI, this work proposes that all of the textural features of the R, G, and B bands and all NDTIs be fused as the input for the winter wheat AGB prediction model. Considering that a large number of image features are mixed with more redundant information, the accuracy of the LR model in estimating AGB may decrease; therefore, the important coefficients of all image features were calculated and sorted to eliminate irrelevant or redundant features, save space, and reduce calculation costs [39]. In the present study, RF was selected to extract data features and rank the importance of features to determine the contribution of each feature to each tree in the RF. Image features whose importance was no less than 10% or 40% of the highest importance were selected as the OTEXS and ONDTIS, respectively. The best texture features and NDTIs were selected

to form the OTEXS and ONDTIS, which were used as the input of the LR model to estimate wheat AGB, thus making for better results from the NDTI [14]. The performance of the AGB model based on the optimal subset of image features was improved at the post-seedling stage and across all stages.

*4.3. Comparison of Regression Methods at Multiple Growth Stages*

PLS regression and RF were selected to establish an AGB prediction model based on fusion data. The prediction model based on the combination of spectral and image features (OTEXS + EWs) had better prediction performance than that based on spectral feature combinations (vegetation index and feature effective wavelength) and image feature combinations (OTEXS and ONDTIS) (Table 9). The combination of image and spectral information provides richer information [40,41]. The RF model performed better at the post-seedling stage because it built a large number of decision trees in the training process while dealing with a large number of samples, thus leading to good performance when dealing with outliers and noise.

**5. Conclusions**

The prediction accuracy of winter wheat AGB can be improved by using a combination feature analysis method. The contribution of image and spectral features and their combination to predict AGB was evaluated. LR, PLS, and RF were used to evaluate the ability of the combination to estimate winter wheat AGB based on the combined feature parameters (OTEXS + VI, OTEXS + EWs, EWs + ONDTI, and VI + ONDTI). It has a great advantage in predicting wheat AGB because the canopy structure information was most accurately captured by digital images. The LR, PLS, and RF models based on the combination of the OTEXS from image features and EWs from spectral features had the highest accuracy in estimating winter wheat AGB at the post-seedling stage ($R^2$ = 0.788 and RMSE = 0.059). Feature selection has great potential to improve the ability of researchers to estimate winter wheat AGB when compared with tradition textural features at the post-seedling stage. This method has a certain reference value for estimating crop AGB at the post-seedling stage. In future work, more spatial effectives of wheat and analysis models will be used to develop a reliable crop AGB estimation model.

**Author Contributions:** L.Z., Q.C. and J.T. conceived and designed the experiments; L.Z. performed the experiment; L.Z. and Q.C. analyzed the data and wrote the original manuscript; J.T., L.H., J.Z., Y.L. and Y.Z. reviewed and revised the manuscript. All authors have read and agreed to the published version of the manuscript.

**Funding:** This research was supported by the National Natural Science Foundation of China under grant 31701323, the Anhui Provincial Key Research and Development Plan (202004a06020032), the Open Foundation of the National Engineering Research Center for Agro-Ecological Big Data Analysis and Application under grant AE202009, and the Natural Science Foundation of Anhui Province under grant 2008085MF184.

**Data Availability Statement:** We are very sorry that the data of this study are private and therefore cannot be made public.

**Acknowledgments:** We would like to thank Dongzhi Wang, Shaojie Chu, Kaixuan Han, Xujin Hu, and Hecai Yuan from Anhui University for their fieldwork and contributions in data collection.

**Conflicts of Interest:** The authors declare no conflict of interest.

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
