# Peer review of "Estimation of Aboveground Biomass for Winter Wheat at the Later Growth Stage by Combining Digital Texture and Spectral Analysis"

_agronomy, doi:10.3390/agronomy13030865_

Round 1
Reviewer 1 Report
1- Abstract needs more improvements to cover study details. Please rephrase it and add some results including statistical values such as R2, RMSE,…etc. for the ML models used.
2- Also English language needs more editing and improvements
3-Figures need to be more clear, please increase the quality
4- ML algorithms need deep description in methods sections as well as adding structure figure for each model
5- I suggest adding Taylor diagram or Radar Charts to show performance for the models developed.
6- In Discussion section, Please make deep discussion with previous works related to this topic, cite more references and show importance of this study to promote your outcomes obtained.
Author Response
Point 1:Abstract needs more improvements to cover study details. Please rephrase it and add some results including statistical values such as R2, RMSE,…etc. for the ML models used.
Response 1:We have revised the summary according to your requirements, specifically in line 25 to 28.
Point 2: Also English language needs more editing and improvements.
Response 2:Thanks for your advice, we have had our manuscript retouched by a professional company.
Point 3:Figures need to be more clear, please increase the quality.
Response 3:According to your suggestion, we have replaced most of the unclear pictures in the manuscript with clearer ones.
Point 4:ML algorithms need deep description in methods sections as well as adding structure figure for each model.
Response 4:The text description about the structure or principle of the model is added in the article. Thank you very much for your suggestions.
Point 5:I suggest adding Taylor diagram or Radar Charts to show performance for the models developed.
Response 5:We have tried these two charts, but the results of our data may not be suitable to be expressed by Taylor diagram or radar chart. Thank you for your suggestion.
Point 6:In Discussion section, Please make deep discussion with previous works related to this topic, cite more references and show importance of this study to promote your outcomes obtained.
Response 6:According to your request, we have revised the discussion section(line 479) and added more references(line 488).
Reviewer 2 Report
The research question is well addressed and the experiments are well described.
One concern I have about the methods applied and their results is that there no explanation why a feature selection method, or even a feature importance measure (RF method has it for example) was not used. This would maybe show a right balance to fuse the features (texture and vegetation/band indices) for a even more precise estimation.
Author Response
Point 1:One concern I have about the methods applied and their results is that there no explanation why a feature selection method, or even a feature importance measure (RF method has it for example) was not used. This would maybe show a right balance to fuse the features (texture and vegetation/band indices) for a even more precise estimation.
Response 1:You have read very carefully and professionally. We did not use RF for feature extraction, but we used other methods to extract features. Spectral features and image features respectively used different methods, as shown below Line 247 mentioned that SPA was used to select the characteristic wavelength of the spectrum; In line 302 and Table 5, by comparing the relationship between texture features of different bands and AGBs, texture features with the best relationship with AGBs can be selected to form the optimal textural feature subset (OTEXS), and AGB prediction model based on OTEXS can be established.
Reviewer 3 Report
Abstract and Introduction are well presented
Request the author to mention all the wheat varieties in the methods section
What was the basis for chosing 40 plots for the study, any connection to standard data collection techniques ?
Figure 10 can be made larger and higher quality
"This result is consistent with the
previous studies that used convolution neural networks to predict biomass-related
shapes." -line 452 can you please provide reference/benchmark comparison?
Author Response
You have read very carefully, and I will answer each of your questions below.
Point 1:What was the basis for chosing 40 plots for the study, any connection to standard data collection techniques ?
Response 1:Because of the need to monitor and study the biomass changes throughout the wheat reproductive period, wheat data were collected for five reproductive periods, so on average, 40 samples need to be collected for each reproductive period, and a total of 200 samples can be obtained for five reproductive periods, which can meet the needs of data analysis.
Point 2:Figure 10 can be made larger and higher quality.
Response 2:We have replaced the higher quality image below line 348, and we have also replaced most of the image with a clearer image.
Point 3:Line 452 can you please provide reference/benchmark comparison?
Response 3:We have added the reference index to line 451.